# Potassium-mediated bacterial chemotactic response

**Chi Zhang\*, Rongjing Zhang\*, Junhua Yuan\***

Hefei National Research Center for Physical Sciences at the Microscale and Department of Physics, University of Science and Technology of China, Hefei, China

**Abstract** Bacteria in biofilms secrete potassium ions to attract free swimming cells. However, the basis of chemotaxis to potassium remains poorly understood. Here, using a microfluidic device, we found that *Escherichia coli* can rapidly accumulate in regions of high potassium concentration on the order of millimoles. Using a bead assay, we measured the dynamic response of individual flagellar motors to stepwise changes in potassium concentration, finding that the response resulted from the chemotaxis signaling pathway. To characterize the chemotactic response to potassium, we measured the dose–response curve and adaptation kinetics via an Förster resonance energy transfer (FRET) assay, finding that the chemotaxis pathway exhibited a sensitive response and fast adaptation to potassium. We further found that the two major chemoreceptors Tar and Tsr respond differently to potassium. Tar receptors exhibit a biphasic response, whereas Tsr receptors respond to potassium as an attractant. These different responses were consistent with the responses of the two receptors to intracellular pH changes. The sensitive response and fast adaptation allow bacteria to sense and localize small changes in potassium concentration. The differential responses of Tar and Tsr receptors to potassium suggest that cells at different growth stages respond differently to potassium and may have different requirements for potassium.

## eLife assessment

In this **important** study, the authors report a novel measurement of the *Escherichia coli* chemotactic response and demonstrate that these bacteria display an attractant response to potassium, which is connected to intracellular pH level. The experimental evidence provided is **convincing** and the work will be of interest to microbiologists studying chemotaxis.

## Introduction

Potassium is an important ion in the normal physiological process of organisms (*Gründling, 2013*). Potassium ions are a major component in establishing the resting membrane potential and thus ensure proper function of the muscles and nerves. Potassium deficiency can affect many biological functions, such as human heartbeat (*Zacchia et al., 2016*) and photosynthesis and respiration of plants (*Li et al., 2020*). In prokaryotes, such as bacteria, potassium ions play a critical role in maintaining the osmotic pressure, pH value and membrane potential of cells (*Ballal et al., 2007*; *Epstein, 2003*). In addition, both the expression of genes and the activities of enzymes are also regulated by potassium ions (*Ali et al., 2017*; *Ballal et al., 2007*; *Epstein, 2003*; *Nanatani et al., 2015*).

Because of its central role in maintaining the normal physiological state of cells, organisms have evolved a series of regulatory systems to control the intracellular concentration of potassium ions, such as Na$^+$/K$^+$ ATP pumps in higher organisms and potassium transport systems in prokaryotes. Most of the studies on potassium have focused on its effects on cell physiology or the dynamics of

**\*For correspondence:**
zhchi@ustc.edu.cn (CZ);
rjzhang@ustc.edu.cn (RZ);
jhyuan@ustc.edu.cn (JY)

**Competing interest:** The authors declare that no competing interests exist.

potassium transport systems. In contrast, studies on the behavioral response to environmental potassium concentration changes and the signaling pathways involved are very limited for bacteria.

Bacteria sense and respond to chemicals via the chemotaxis signaling pathway. In *Escherichia coli*, chemo stimuli are detected by transmembrane receptors (*Bray et al., 1998*; *Endres et al., 2008*; *Homma et al., 2004*). The sensed signal is then transmitted to the associated cytoplasmic histidine kinase CheA, affecting its autophosphorylation (*Li and Hazelbauer, 2011*; *Li et al., 2013*). CheA transfers its phosphoryl group to the response regulator CheY and the methylesterase CheB, yielding CheY-P and CheB-P, respectively (*Bren and Eisenbach, 2000*). CheY-P binds to the base of the flagellar motor, increasing the probability of motor rotating clockwise (CW), namely motor CW bias, and thus increasing the cell tumble frequency (*Dyer and Dahlquist, 2006*; *Lam et al., 2013*; *van Albada and Ten Wolde, 2009*). The phosphatase CheZ binds to CheY-P and accelerates its dephosphorylation. CheB-P and CheR demethylate and methylate the receptors, respectively, to accomplish robust adaptation in chemotaxis (*Djordjevic and Stock, 1997*; *Djordjevic and Stock, 1998*; *Levin et al., 2002*; *Lupas and Stock, 1989*; *Lybarger and Maddock, 1999*).

Conventional stimuli are sensed by *E. coli* directly or indirectly. The former suggests that the ligand can bind to the periplasmic domain of the corresponding receptor directly and modify kinase activity, such as MeAsp and serine (*Neumann et al., 2010*; *Tajima et al., 2011*). The latter requires the help of binding proteins, such as maltose sensed by the Tar receptor with the help of periplasmic maltose-binding protein (*Brass and Manson, 1984*), and peptides sensed by the Tap receptor with the help of peptide-binding proteins (*Manson et al., 1986*). However, the detailed mechanism of bacterial sensing and response to ionic stimuli, such as the chemotactic repellent, nickel ions, is not clear (*de Pina et al., 1995*; *Englert et al., 2010*). The chemotactic response of other ions also remains to be studied.

It was discovered recently that bacteria in biofilms secrete potassium into their surroundings through ion channels on the cell membrane, thereby affecting the behavior of distant free bacteria and attracting them (*Humphries et al., 2017*; *Prindle et al., 2015*). However, how limited changes in potassium concentration regulate the movement of bacteria, especially after long-distance diffusion, is still unclear.

Here, we constructed a linear concentration gradient of potassium ions with a microfluidic device, and found obvious taxis to the region of high concentration for wild-type *E. coli*. To determine the origin of this taxis, we measured the response of motor speed and CW bias to a stepwise increase in potassium concentration. The motor CW bias exhibits an attractant-like response, while the motor speed remains constant, which suggests a constant proton motive force (PMF). The chemotaxis-defective strain did not respond to this stimulus. This confirmed that the CW bias response of motors resulted from the upstream chemotaxis signaling pathway. To further characterize the response of the chemotactic signal to potassium, we directly measured the response of kinase activity to different concentrations of potassium chloride by monitoring the Förster resonance energy transfer (FRET) between CheY-eYFP (yellow fluorescent protein) and CheZ-eCFP (cyan fluorescent protein). We ruled out the possibility of osmotaxis by comparing the responses to 30 mM potassium chloride and 60 mM sucrose. The dose–response curve and step response of kinase activity to potassium suggested a sensitive sensing process and fast adaptation. Further experiments with mutants expressing Tar, Tsr, or no receptor suggested that this chemotactic response may result from the increase in intracellular pH. Our findings suggest a new mechanism of chemotactic response to ionic stimuli. Employing a coarse-grained chemotaxis model with parameters extracted from our measurements, we performed stochastic simulation of *E. coli* cells in a potassium gradient generated by a biofilm producing an oscillating potassium signal, and demonstrated the delayed periodic attraction of the cells to the biofilm.

## Results

### Chemotaxis in a linear concentration gradient of potassium

It has been reported that potassium ions work as a communication agent in swarms and biofilms for both *Bacillus subtilis* and *Pseudomonas aeruginosa*. A high concentration of potassium chloride (300 mM) could attract swimming cells (*Humphries et al., 2017*; *Prindle et al., 2015*). To study the chemotactic response of *E. coli* to potassium, we set up a linear concentration profile of potassium

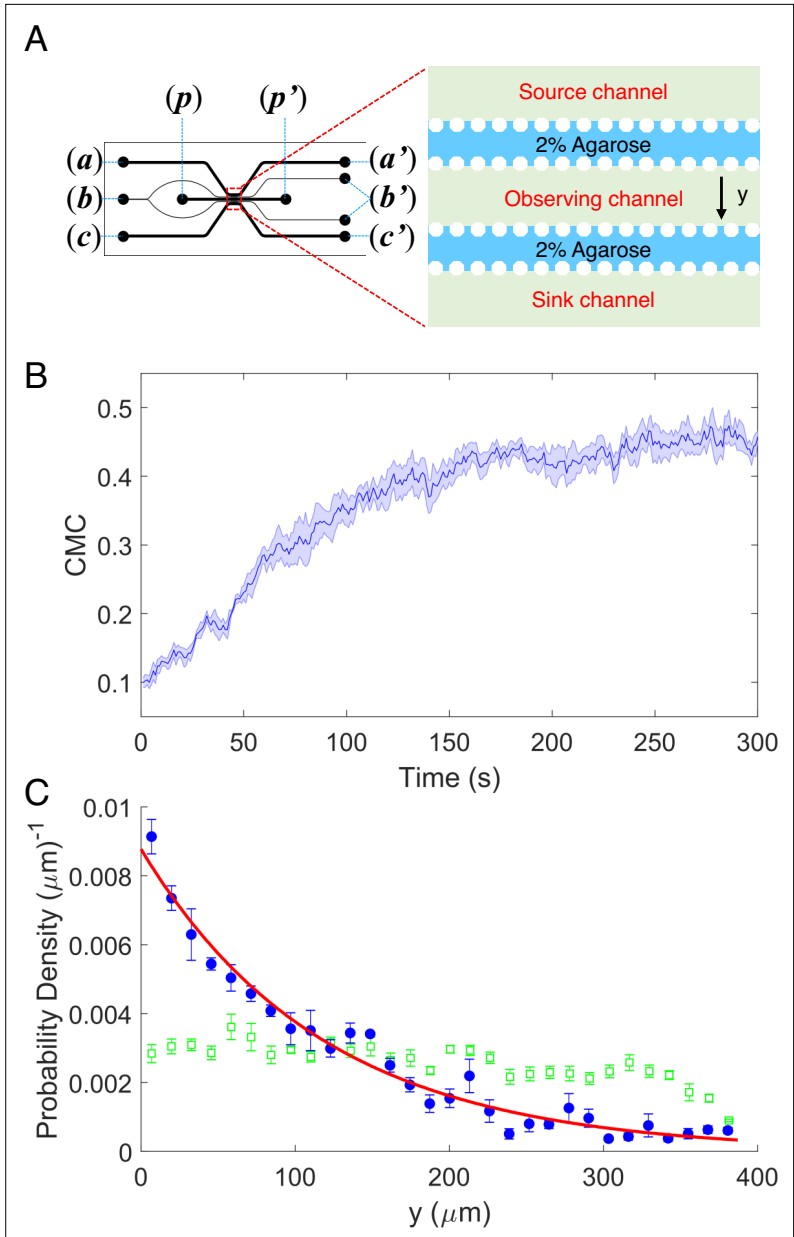

**Figure 1.** The chemotaxis performance of *E.coli* in a linear concentration gradient of potassium. (**A**) Design diagram of the microfluidic device. The source channel and sink were flowed with 100 mM KCl and motility buffer, respectively. The inlets of KCl, agarose, motility buffer, and cells are denoted by (*a*), (*b*), (*c*), and (*p*), respectively. The outlets are labeled by the corresponding letters with the prime symbol. (**B**) The average chemotaxis migration coefficient (CMC) of four datasets as a function of time for the wild-type strain (HCB1) under a linear concentration gradient of KCl. The shaded area denotes standard error of the mean (SEM). (**C**) The cell density profile in the observing channel along the *y*-axis at the beginning (*t* = 1 s, green squares) and a steady state (*t* = 300 s, blue dots). The red solid line is an exponential fit to the data. Error bars denote SEM.

chloride with a microfluidic device designed as described previously (*Liu et al., 2022*; *Tian et al., 2021*).

As shown in *Figure 1A* a five-channel diffusion device was constructed. 2% (wt/vol) agarose was injected into the two resistance channels through port (*b*) to block the passage of cells without influencing the diffusion of potassium. A 100-mM potassium chloride solution, prepared in the potassium-depleted motility buffer, flows through the source channel from port (*a*) to (*a'*), while potassium-depleted motility buffer flows through the sink channel from port (*c*) to (*c'*). Potassium diffuses from the source

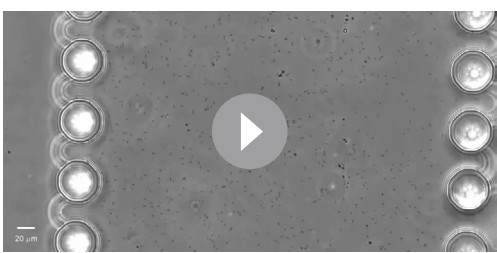

**Video 1.** An example video of wild-type *E. coli* HCB1 cells swimming up the potassium gradient in a microfludic device. The left side is the source channel. Scale bar = 20 μm.

https://elifesciences.org/articles/91452/figures#video1

channel to the sink channel and forms a linear concentration gradient in the observation channel (*Tian et al., 2021*). The wild-type strain HCB1 was sealed in the observation channel through port (**p**) with a drop of hot agarose (2% wt/vol). The movements of the cells were recorded by a high-speed CMOS camera. Four movies were recorded. An example video is shown in *Video 1*. We calculated the displacement of the mean cell position for each movie. The average results are plotted in *Figure 1B*. The shaded area denotes standard error of the mean (SEM). We also computed the cell density profile along the *y*-axis in the observation channel at the initial time of *t* = 1 s and steady state (*t* = 300 s). As shown in *Figure 1C*, the green squares and blue dots are experimental data. The error bars denote SEM. We found that the distribution of cells is exponential at steady state, similar to the results of logarithmic chemotactic sensing in a linear MeAsp gradient (*Kalinin et al., 2009*). The red solid line denotes the exponential fit, with a fitted decay constant of 0.0085 ± 0.0004 μm$^{-1}$. As reported previously (*Kalinin et al., 2009*), the fitted exponential decay constant equals $v_d/v_s$, where $v_d$ and $v_s$ denote the drift velocity and cell motility constant, respectively. For wild-type *E. coli* with $v_s$ = 53.2 μm$^2$/s (*Wu et al., 2006*), we obtained a drift velocity $v_d$ = 0.45 ± 0.02 μm/s.

According to our results, the wild-type strain exhibited significant movement toward the area of high concentration in a linear concentration gradient of potassium chloride, which was similar to its chemotaxis in a typical attractant concentration gradient (*Jiang et al., 2010*; *Kalinin et al., 2009*; *Liu et al., 2022*; *Son et al., 2016*). Thus, *E. coli* could be attracted by potassium and perform a trend movement in a linear gradient field on the order of millimoles. We sought to further investigate the characteristics and mechanisms of this attractant-like response.

## The response of the motor rotational signal to potassium

Potassium is important for maintaining cell membrane potential, which is one of the two components of PMF. PMF is the energy source for the flagellar motor. The absolute value of the transmembrane electrical potential will decrease when the concentration of extracellular potassium increases. This may lead to a change in PMF and affect bacterial chemotaxis by directly changing the motor behavior. However, earlier studies found that a sudden change in membrane potential in *E. coli* would result in a reverse change in the transmembrane proton concentration difference, thus keeping PMF virtually constant (*Bakker and Mangerich, 1981*).

To test whether the chemotactic response of *E. coli* to potassium resulted from possible PMF changes, we monitored the motor response of the wild-type strain (JY26-pKAF131) to a stepwise addition of 30 mM potassium chloride using a bead assay. As shown in *Figure 2A*, the cell body was attached to a 0.01% poly-L-lysine-coated cover slide. A 1-μm-diameter bead was marked on the truncated filament. We used a high-speed CMOS camera to record the rotation of the bead. As shown in *Figure 2B*, the typical traces of motor rotational speed (blue line) and CW bias (purple line) were calculated. The positive and negative values of speed denote CCW (counter-clockwise) and CW rotation, respectively. The CW bias is positively correlated with the intracellular concentration of CheY-P (*Cluzel et al., 2000*), and the rotational speed is proportional to the PMF (*Gabel and Berg, 2003*). The stimulus (30 mM KCl) was added at *t* = 120 s and removed at *t* = 480 s. The average trace of 83 motors is shown in *Figure 2C*. The moments of addition and removal of stimulus were shifted to *t* = 0, which are marked by the purple dashed line and green dashed line, respectively. The CW bias decreased upon addition of potassium chloride, and then it adapted to a higher level than the pre-stimulus value. Such over adaptation was similar to the response of CW bias to sucrose in osmotaxis (*Rosko et al., 2017*). However, the motor speed remained constant during this process, confirming that the PMF remained constant, which was different from the osmotaxis response of motors. The results suggested that the response of motor CW bias to potassium was not due to PMF change. We performed further experiments with the chemotaxis-defective strain (HCB901-pBES38). The average

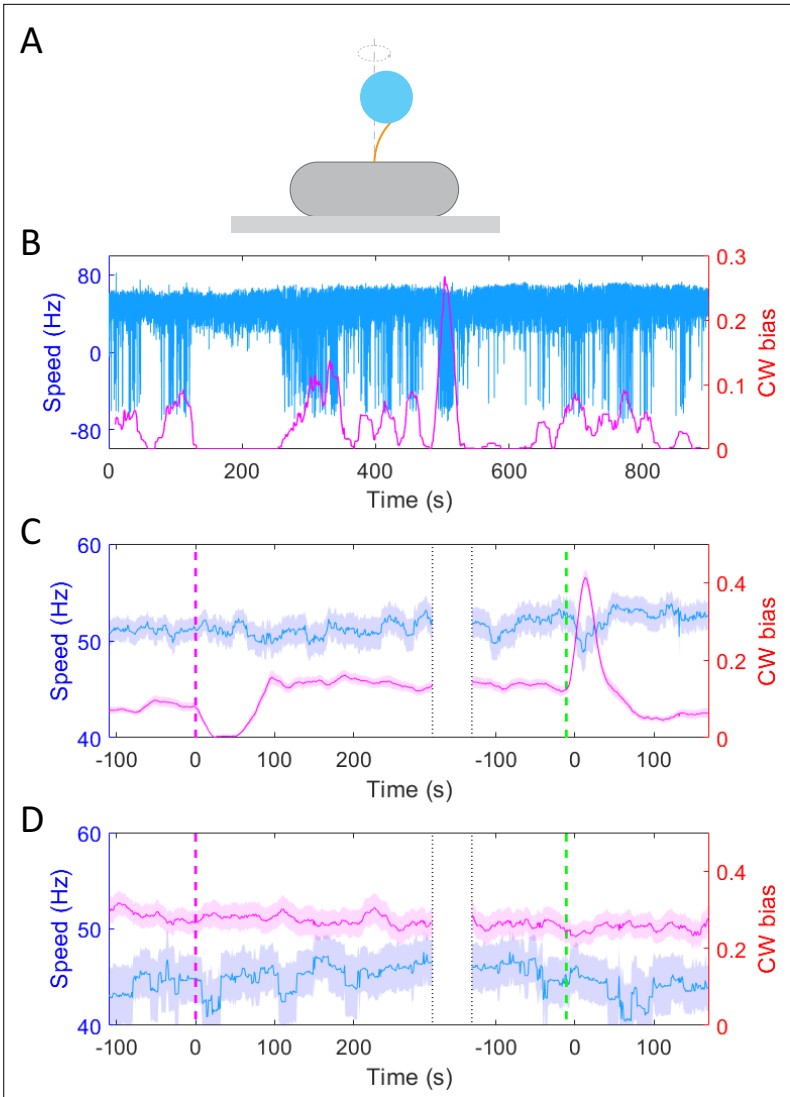

**Figure 2.** The response of motor rotational signal to potassium. (**A**) Schematic diagram of the bead assay for the flagellar motor. (**B**) Typical trace of rotational speed (blue line) and clockwise (CW) bias (purple line) of individual motors for the wild-type strain (JY26-pKAF131). The positive and negative values of speed denote counter-clockwise (CCW) and CW rotation, respectively. 30 mM KCl was added at $t = 120$ s and removed at $t = 480$ s. (**C**) The average response of 83 motors from 5 samples for the wild-type strain to 30 mM KCl. The vertical purple (green) dashed lines indicate the moment of adding (removing) stimulus. The shaded areas denote standard error of the mean (SEM). (**D**) The average response of 22 motors from 4 samples for the chemotaxis-defective strain (HCB901-pBES38) to 30 mM KCl. The vertical purple (green) dashed lines indicate the moment of adding (removing) stimulus. The shaded areas denote SEM.

The online version of this article includes the following figure supplement(s) for figure 2:

**Figure supplement 1.** The response of motor clockwise (CW) bias to 15 mM $K_2SO_4$ (red line) and 30 mM KCl (blue line) for the wild-type strain.

result of 22 motors exhibited no response to 30 mM potassium chloride for either CW bias or speed (*Figure 2D*), especially on the time scale of the chemotactic response as seen in the wild-type cells. These results confirmed that the CW bias response of the motor to potassium resulted from the upstream chemotaxis signaling pathway.

## The response of the chemotactic signal to potassium

To directly measure the response of the chemotactic signal to potassium ions, we monitored the receptor-kinase activity in vivo by monitoring the FRET between CheY-eYFP and CheZ-eCFP.

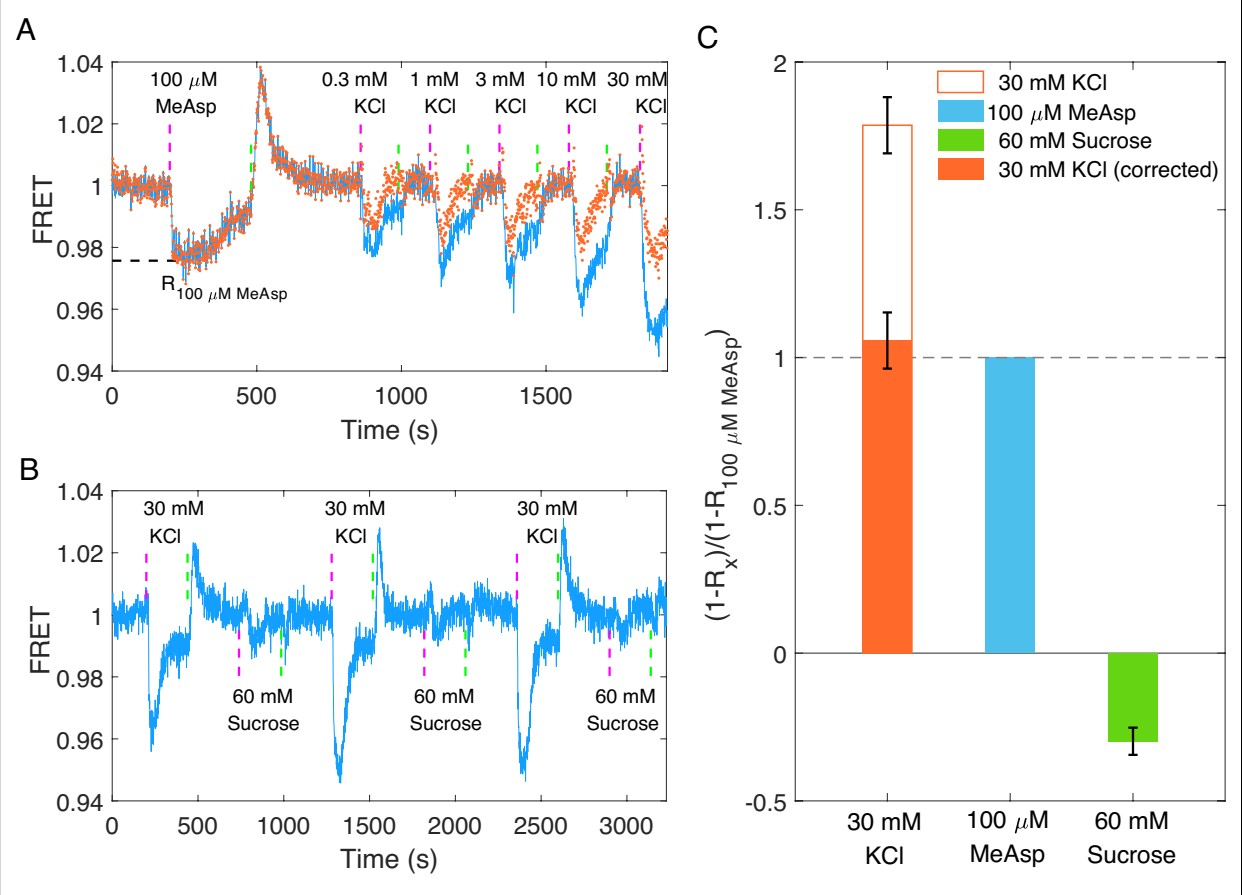

**Figure 3.** The chemotactic response of the wild-type strain (HCB1288-pVS88) to potassium. (**A**) Chemotactic response of the wild-type strain (HCB1288-pVS88) to stepwise addition and removal of KCl. The blue solid line denotes the orignal signal, and the red dots represent the pH-corrected signal, which was recalculated from the pH-corrected cyan fluorescent protein (CFP) and yellow fluorescent protein (YFP) channels using the response of the no-receptor strain. (**B**) Comparison of the chemotactic response to 30 mM KCl and 60 mM sucrose. The vertical purple (green) dashed lines indicate the moment of adding (removing) stimulus. (**C**) Quantitative comparison among the responses to 100 µM MeAsp, 30 mM KCl, and 60 mM sucrose. The hollow and solid red bars represent the value calculated from the original signal and the pH-coPlrrected signal, respectively. The errors denote standard error of the mean (SEM).

The online version of this article includes the following figure supplement(s) for figure 3:

**Figure supplement 1.** Quantitative comparison of the response of the chemotactic signal to 10 mM KCl and 5 mM K₂SO₄.

Using the FRET setup described previously (*Zhang et al., 2018*), we measured the chemotactic response of a cell population with the wild-type strain (HCB1288-pVS88). We measured the response of the same sample to potassium chloride at different concentrations and to 100 µM MeAsp (a typical saturated attractant for the Tar receptor in *E. coli*). As shown in *Figure 3A*, potassium (blue solid line) induces a significant chemotactic response as an attractant. Compared with 100 µM MeAsp, 30 mM KCl resulted in a larger response and faster adaptation. Moreover, the chemotactic signal exhibits an imprecise adaptation. This is different from the over adaptation exhibited by motor CW bias. To exclude the effect of chloride ion, we quantitatively compared the response of the chemotactic signal to 10 mM KCl and 5 mM K₂SO₄, both of which contain 10 mM potassium ion with or without chloride ion. As shown in *Figure 3—figure supplement 1*, they induce similar responses. We also measured the response of the motor rotational signal to 15 mM K₂SO₄ using the bead assay and compared it with the response to 30 mM KCl (*Figure 2C*). The results are shown in *Figure 2—figure supplement 1*. The response of CW bias to 15 mM K₂SO₄ exhibited an attractant response, characterized by a decreased CW bias upon the addition of K₂SO₄, followed by an over-adaptation that is qualitatively similar to the response to 30 mM KCl. Thus, the chemotactic response mainly resulted from potassium ions rather than chloride ions.

Earlier studies have shown that changes in osmotic pressure caused by a change in ion concentration on the millimole scale can also lead to chemotactic responses in *E. coli* by altering the interaction between receptors (*Vaknin and Berg, 2006*). Here, we obtained a similar over adaptation in the CW bias response of motors as that of osmotaxis. To determine whether the chemotactic response to potassium was due to changes in osmotic pressure, we compared the response to both 30 mM KCl and 60 mM sucrose, both of which induced the same osmotic pressure. The results are shown in *Figure 3B*. In our measurements, 60 mM sucrose induced a repellent-like response, which is consistent with previous work (*Vaknin and Berg, 2006*), whereas 30 mM KCl induced an attractant response. Thus, the chemotactic response of *E. coli* to KCl did not result from osmotaxis. The quantitative comparison of the three types of chemicals is shown in *Figure 3C*. The response to 30 mM KCl was 1.79 ± 0.10 times as large as that to 100 μM MeAsp, while the response to 60 mM sucrose was −0.30 ± 0.05 times as large as that to 100 μM MeAsp.

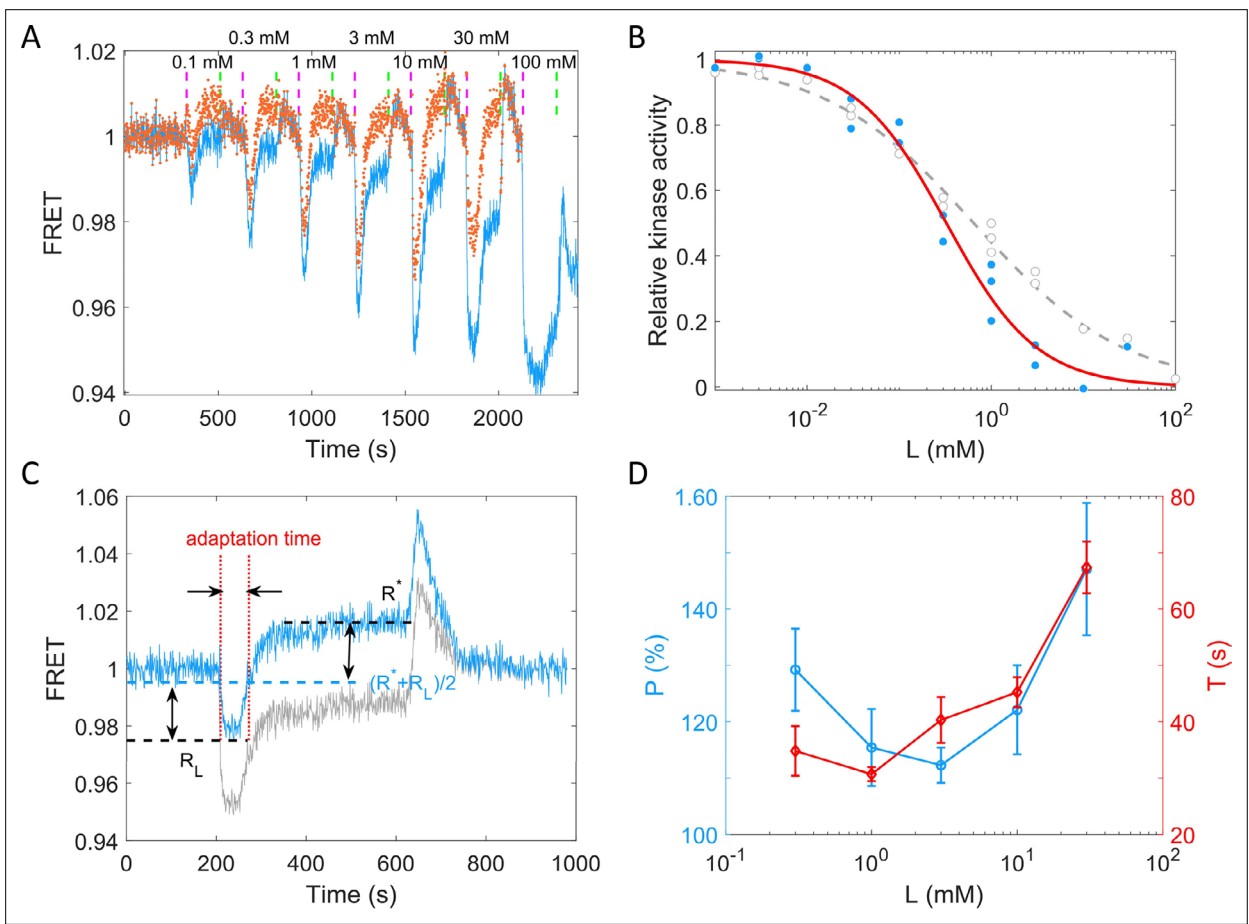

**Figure 4.** Quantitative results of the chemotactic response of the wild-type strain (HCB1288-pVS88) to potassium. (**A**) A typical example of the dose–response measurement. The blue solid line denotes the original signal, and the red dots represent the pH-corrected signal, recalculated from the pH-corrected cyan fluorescent protein (CFP) and yellow fluorescent protein (YFP) signals using the response of the no-receptor strain. The vertical purple (green) dashed lines indicate the moment of adding (removing) stimulus. (**B**) The dose–response curve of relative kinase activity to KCl. The blue dots and gray circles represent the pH-corrected and original experimental data, respectively. The red solid and gray dashed lines are the fit curves for the blue dots and gray circles, respectively, using a Hill function. The fitted Hill coefficient for original and pH-corrected response were 0.53 ± 0.04 and 0.88 ± 0.14, respectively, and the concentration for half-maximal response ($K_{0.5}$) were 0.64 ± 0.12 and 0.33 ± 0.06 mM, accordingly. (**C**) Definition of adaptation time in the step response. The gray and blue lines represent the original and pH-corrected signals, respectively. (**D**) The adaptation level $P = (R^* - R_L) / (1 - R_L)$ and adaptation time ($T$) as a function of the concentration of KCl, calculated with pH-corrected data. The errors denote standard error of the mean (SEM).

The online version of this article includes the following figure supplement(s) for figure 4:

**Figure supplement 1.** The response of wild-type strain (HCB1288-pVS88) to KCl.

## Quantitative characterization of the chemotactic response to potassium

To quantitively describe the response of receptor-kinase activity to potassium ions, we measured the dose–response curve of HCB1288-pVS88 to potassium chloride. As shown in *Figure 4A*, different concentrations of KCl were added and then removed. The FRET value of the immediate response after adding each concentration of stimulus was recorded. The FRET values were normalized by the pre-stimulus FRET value. The relative kinase activity was obtained by rescaling the FRET values of 1 (the pre-stimulus value) to 0.94 (the value after adding a saturated concentration of stimulus) to the range between 1 and 0, then the relation between relative kinase activity and concentrations of KCl was obtained (*Figure 4B*).

As shown in *Figure 4B*, the gray circles represent experimental data for potassium. We fitted them with a Hill function (gray dashed line). The fitted Hill coefficient was 0.53 ± 0.04. However, this value was revised to 0.88 ± 0.14 when we corrected the FRET responses for the pH effects on the brightness of eCFP and eYFP (the blue dots and red solid line). The concentration for half-maximal response ($K_{0.5}$) was 0.64 ± 0.12 mM before correction and 0.33 ± 0.06 mM after correction. To study the adaptation kinetics, we measured the step response for different concentrations of potassium. As shown in *Figure 4C*, we calculated the adaptation percentage $P = \left(R^* - R_L\right) / \left(1 - R_L\right)$ and adaptation time $T$, where $R^*$ and $R_L$ are the adapted value and the lowest value of the FRET signal after stepwise addition of potassium, respectively. The adaptation percentage ($P$) and the adaptation time ($T$) for different concentrations of potassium are shown in *Figure 4D*.

According to our measurements, *E. coli* responds sensitively (with a small $K_{0.5}$) and adapts quickly to potassium chloride in a range of 0.01–100 mM. The recovery of the chemotactic response was imprecise for all concentrations of potassium ions.

## The chemotactic response to potassium may result from the intracellular pH increase

We showed that the attractant-like response of the chemotactic signal to potassium does not result from osmotaxis. Moreover, the PMF or motor speed did not change during that process. Considering that the absolute value of the transmembrane potential deceases as the concentration of extracellular potassium ions increases, the intracellular pH will increase to maintain a constant PMF. To confirm this, we monitored the intracellular pH using the pH-sensitive fluorescent protein pHlourin2, by computing the ratio of emitted fluorescence intensities with 405 and 488 nm excitations (the ratio increases with pH value) (*Mahon, 2011*). The results are shown in *Figure 5A*. The intracellular pH of the wild-type strain (HCB33-pTrc99a_pHlourin2) increases immediately upon addition of 30 mM KCl. The purple arrows denote the moment of adding stimulus, while green arrows denote the moment of removing stimulus. As a control, 20 mM sodium benzoate with pH = 4.55 was added at $t$ = 720 s to decrease the intracellular pH to 4.55 (*Nakamura et al., 2009*).

It was reported that Tar and Tsr mediate opposite responses to the changes in intracellular pH (*Krikos et al., 1985*; *Umemura et al., 2002*). Therefore, we measured the chemotactic response of strains expressing only the Tar or Tsr receptor via the FRET assay. As shown in *Figure 5B*, four different concentrations of KCl were added and removed successively, and the kinases activity of the Tsr-only strain exhibited an attractant response. Its imprecise adaptation is apparent as the potassium concentration increases. In contrast, the Tar-only strain exhibited a biphasic response, with kinase activity increasing sharply and then decreasing rapidly below the pre-stimulus level upon addition of potassium (*Figure 5C*). Such a biphasic response was also observed in the response of the Tar-only strain to 40 mM sodium benzoate at pH of 7.0 (*Figure 5—figure supplement 1*). Note that the cytoplasmic pH deceases upon addition of sodium benzoate at pH of 7.0, so that the response to addition of sodium benzoate is opposite to the response to addition of potassium for the Tar-only strain. Therefore, the different responses of the Tar- and Tsr-only strains to potassium were consistent with the responses to the increase in intracellular pH.

We noted that both major receptors exhibited imprecise adaptation to stepwise addition of potassium. To investigate whether this was due to chemotactic sensing, we monitored the response of the strain with no chemoreceptors to stepwise addition of potassium. Upon addition of potassium, the FRET value decreased and remained at a constant value similar to the adapted level for the strains with chemoreceptors (*Figure 5D*). This indicated that the imprecise adaptation seen for the wild-type, Tar-only, and Tsr-only strains was not due to chemosensing. As the solution pH affects the brightness

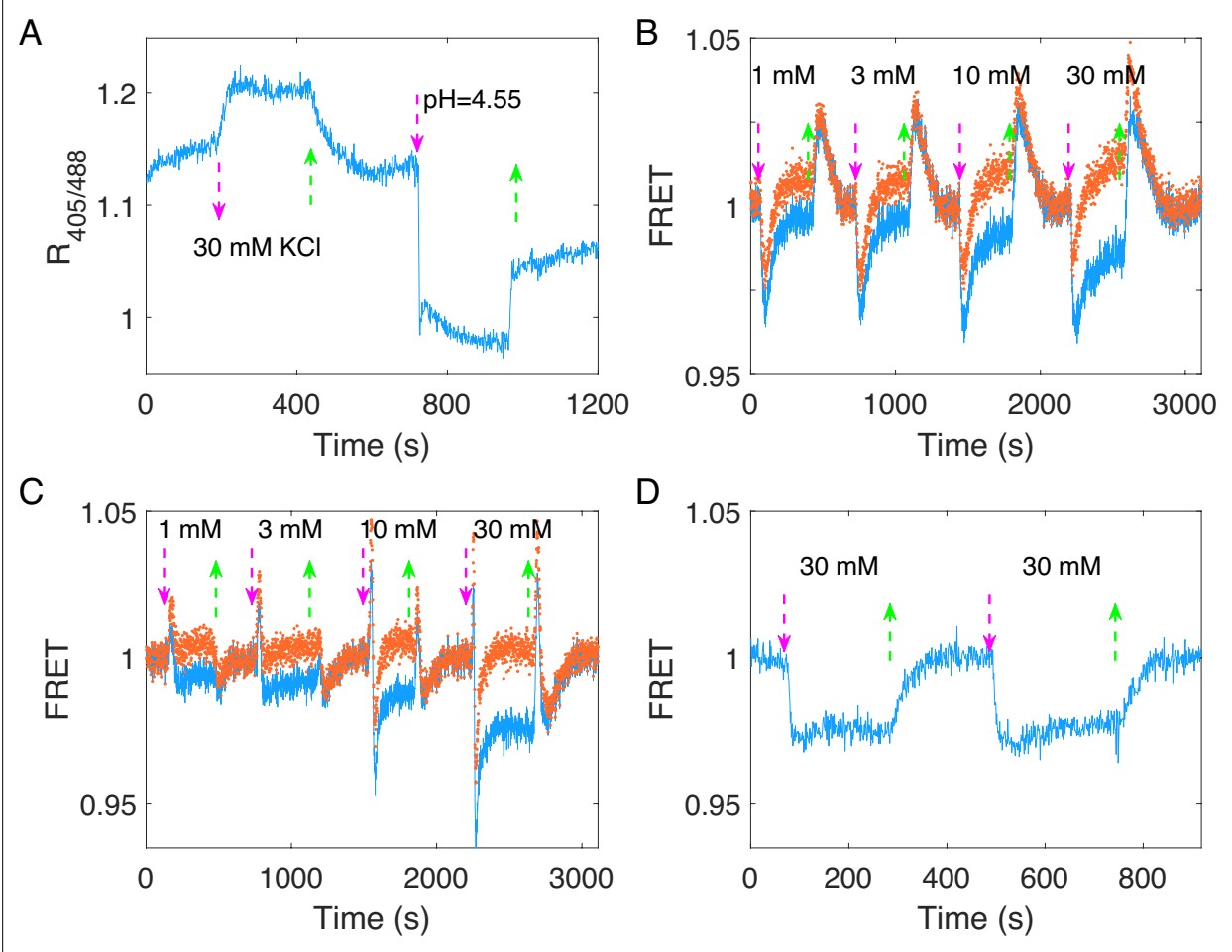

**Figure 5.** The response of intracellular pH for the wild-type strain and the chemotactic response of different mutant strains to potassium. (**A**) The response of intracellular pH to 30 mM KCl for the wild-type strain (HCB33-pTrc99a_pHluorin2). The response to 20 mM sodium benzoate solution with pH = 4.55 was used as a control. (**B**) The chemotactic response of the Tsr-only strain (HCB1414-pPA114-pVS88) to four typical concentrations of potassium. The blue line denotes the orignal signal, and the red dots represent the pH-correcting signal. (**C**) The chemotactic response of the Tar-only strain (HCB1414-pLC113-pVS88) to four typical concentrations of potassium. The blue line denotes the orignal signal, and the red dots represent the pH-correcting signal. (**D**) The chemotactic response of the no-receptor strain (HCB1414-pVS88) to 30 mM KCl. The vertical purple (green) arrows denote the moment of adding (removing) stimulus.

The online version of this article includes the following figure supplement(s) for figure 5:

**Figure supplement 1.** The chemotactic response of the Tar-only strain (HCB1414-pLC113-pVS88) to 40 mM sodium benzoate at pH = 7.0.

**Figure supplement 2.** The cyan fluorescent protein (CFP) intensity response to 30 mM KCl.

**Figure supplement 3.** The response of the no-receptor mutant (HCB1414-pVS88) to different concentrations of KCl.

of fluorescent proteins, we hypothesize that this apparent imprecise adaptation was due to the differential change in eCFP and eYFP brightness when the cytoplasmic pH changed. To test this hypothesis, we bleached the eYFPs of the no-receptor strain (HCB1414-pVS88), eliminating possible FRET between CheY-eYFP and CheZ-eCFP, and compared the response of CFP fluorescence intensity to 30 mM KCl before and after bleaching. As shown in *Figure 5—figure supplement 2A*, the CFP intensity showed similar amount of increase upon addition of KCl (i.e., increase of cytoplasmic pH) before and after YFP bleaching. Thus, such a response did not result from FRET, but from changes in fluorescence intensity due to an increase in intracellular pH. We calculated the ratio of CFP fluorescence intensity $P = F_r/F_b = 1.056$ without FRET, where $F_r$ and $F_b$ were the CFP intensity for cells in 30 mM KCl and motility medium, respectively. From this ratio, we could calculate the theoretical enhancement in CFP intensity due to pH increase for the wild-type strain (HCB1288-pVS88) after adding 30 mM KCl (red dashed line in *Figure 5—figure supplement 2B*), which we found was similar to the adapted level

of the CFP signal. This further proved that the apparent imprecise adaptation in the FRET signal was due to changes in eCFP and eYFP brightness when the cytoplasmic pH changed.

## Revised FRET responses by correcting the pH effects on the brightness of eCFP and eYFP

To minimize the impact of pH changes altering the fluorescent protein brightness on FRET measurements of chemotactic response and adaptation to potassium, we measured the full potassium response curve for the no-receptor mutant (HCB1414-pVS88), as shown in *Figure 5—figure supplement 3*. We characterized the pH effects on CFP and YFP channels at different concentrations of KCl, and the relationship between the ratio of the signal post- to pre-KCl addition and the KCl concentration was established for both channels, as shown in *Figure 5—figure supplement 3C*. The pH-corrected signal after KCl addition for strains with receptors was obtained by dividing the original signal after KCl addition by this ratio at the specific KCl concentration. This procedure was applied to both CFP and YFP channels. The pH-corrected responses for the Tar- and Tsr-only strains are represented by red dots in *Figure 5BC*.

We recalculated the FRET responses to stepwise addition of KCl, with an example depicted by the red dots in *Figure 3A*. The corrected response magnitude to 30 mM KCl is similar to that of 100 μM MeAsp, being 1.06 ± 0.10 times as large, as shown by the red bar in *Figure 3C*. We also calculated the dose–response curve and the adaptation curve from the pH-corrected signals, as shown in *Figure 4*. The FRET signal also exhibits over-adaptation, similar to the bead assay, when we recalculated the response by correcting the CFP and YFP channels. The fitted Hill coefficient for the dose–response curve is 0.88 ± 0.14 (mean ± standard deviation [SD]), which is close to the response to MeAsp (~1.2) (*Sourjik and Berg, 2002b*), and the concentration for half-maximal response ($K_{0.5}$) was 0.33 ± 0.06 mM (mean ± SD).

Our results indicated that the increase in the extracellular concentration of potassium leads to an increase in the intracellular pH. The Tar and Tsr receptors respond to the increase in intracellular pH differently. The wild-type strain exhibits a similar response to the Tsr-only strain due to the larger amount of Tsr than Tar receptors. These responses could be adapted via methylation and demethylation of receptors.

## The chemotaxis-based simulation of *E. coli* attraction by the potassium signal produced by a biofilm

Based on the mechanism of potassium sensing we established here, we sought to simulate the chemotactic swimming of *E. coli* cells in response to a periodic potassium signal secreted from a typical biofilm (*Humphries et al., 2017*).

To simulate the periodically fluctuating field of potassium concentration produced by the biofilm, an oscillating source was introduced using a cosine function $L_s = L_0 \left(1 - \cos\left(2\pi t/T\right)\right)$ at position $x = 0$, where $L_0$ is the half maximum concentration of potassium, and $T$ denotes the period of oscillation. In a semi-infinite liquid environment, this source results in an oscillating spatiotemporal profile of potassium

$$L\left(x, t\right) = L_0 \left[1 - \cos\left(\frac{2\pi t}{T} - \sqrt{\frac{\pi}{DT}}x\right) \exp\left(-\sqrt{\frac{\pi}{DT}}x\right)\right], \qquad (1)$$

where $D = 1333.3 \ \mu\text{m}^2/\text{s}$ is the diffusion constant of potassium in water (*Fell and Hutchison, 1971*). The profile is shown in *Figure 6A* (see Appendix 1 for the detailed derivation).

Similar to the two-state model of pH taxis of *E. coli* (*Hu and Tu, 2013*; *Hu and Tu, 2014*), we employed a coarse-grained model of the chemotaxis signaling pathway to simulate the chemotactic motion of *E. coli* in the potassium profile (see Materials and methods for details) (*Liu et al., 2022*; *Tian et al., 2021*). We utilized the Monod–Wyman–Changeux allosteric model (*Monod et al., 1965*) to describe the potassium sensing by chemoreceptors (*Tu et al., 2008*):

$$a = \frac{1}{1 + \exp\left[N\left(f_m\left(m\right) + f_L\left(L\right)\right)\right]}, \qquad (2)$$

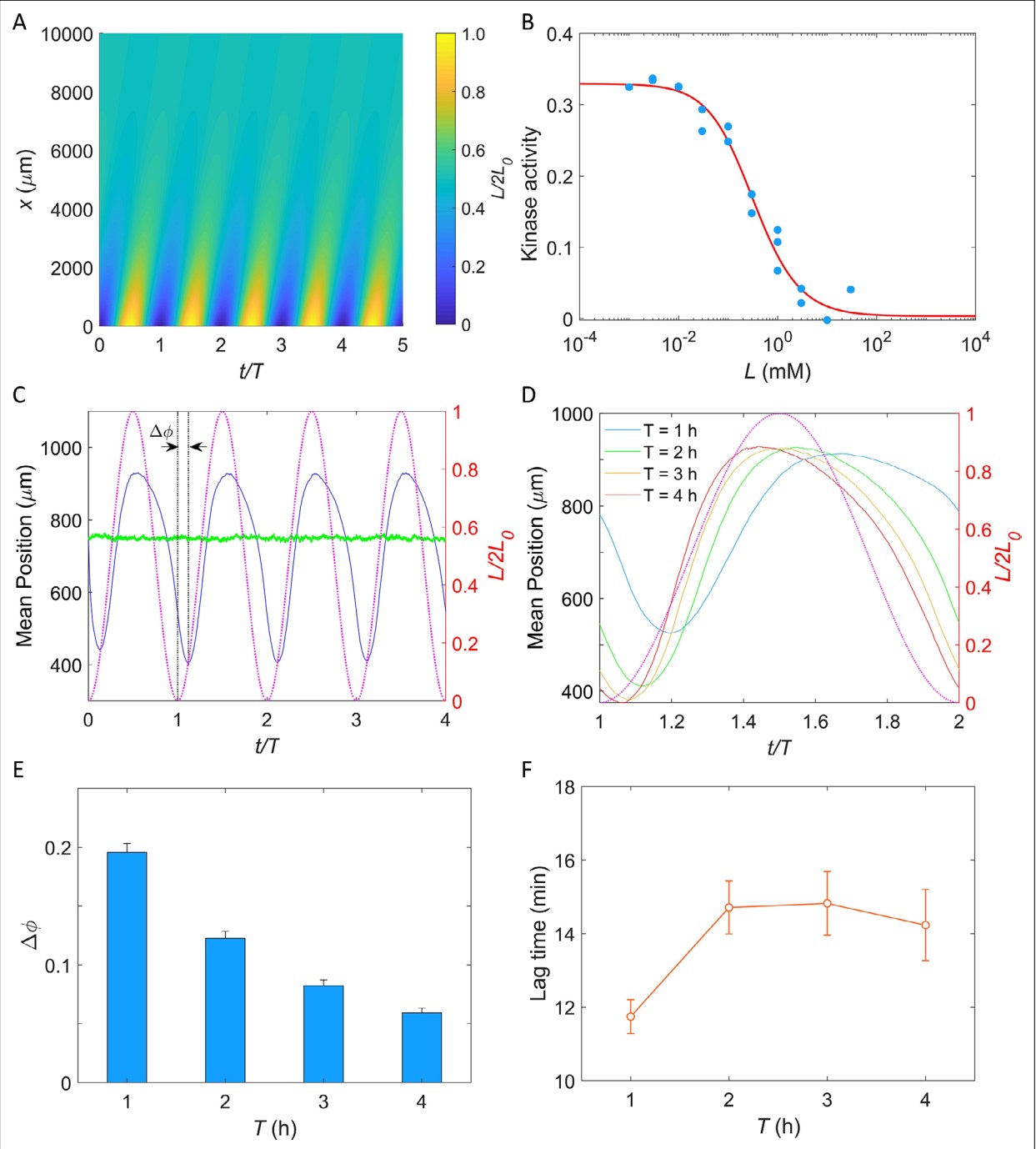

**Figure 6.** Simulation of *E. coli* chemotactic swimming in response to a periodic potassium signal produced by a typical biofilm. (**A**) The oscillating spatial gradient of potassium. (**B**) The dose–response curve of receptor-kinase activity to potassium. Blue dots are pH-corrected experimental data (blue dots in *Figure 4B*). Red solid line is the fitting curve with *Equation 2*. (**C**) Typical traces of simulated mean positions with $L_0 = 1.0$ mM and $T = 2$ hr. The blue and green lines denote the mean $x$-position ($1500 - \langle x \rangle$) and mean $y$-position ($\langle y \rangle$), respectively. The purple dashed line indicates the oscillating potassium source at $x = 0$. The phase delay $\Delta\phi$ is defined as the phase shift between the trough of the mean $x$-position and the trough of potassium source except for $t/T = 0$. (**D**) The comparison of the mean $x$-position under different periods of the driving source: $T = 1$, 2, 3, and 4 hr. The purple dashed line indicates the oscillating potassium source at $x = 0$. (**E**) The relation between the phase delay $\Delta\phi$ and the driving period $T$. Each data was calculated by the average of 10 simulations. The error denotes standard deviation. (**F**) The relation between lag time ($\Delta\phi \cdot T$) and the driving period $T$. The error denotes standard deviation.

The online version of this article includes the following figure supplement(s) for figure 6:

**Figure supplement 1.** The relation between phase delay ($\Delta\phi$) and driving periods ($T$) with different $L_0$.

**Figure supplement 2.** The relationship between the lag time ($\Delta\phi \cdot T$) and the methylation rate $k_R$.

where $a$ is the kinase activity of the cluster of chemoreceptors, $m$ is the methylation level of receptors, $L$ denotes the concentration of extracellular potassium, $N$ is the number of receptor homodimers in an allosteric cluster, and $f_m$ and $f_L$ represent the methylation- and ligand-dependent free energy, respectively.

$$f_L\left(L\right) = \ln\frac{1 + L/K_{off}}{1 + L/K_{on}},$$

$$f_m\left(m\right) = \alpha\left(m - m_0\right),$$

where $K_{off}$ and $K_{on}$ are the ligand dissociation constants for inactive and active receptors, respectively, $\alpha$ is the free energy change per added methyl group, and $m_0$ is the methylation level where $f_m$ crosses zero. As shown in *Figure 6B*, the dose–response of kinase activity to potassium could be well fitted by *Equation 2* to extract the parameters.

In our simulation, cells swim in a square space with dimensions of $0 \leq x \leq 1500\,\mu$m and $0 \leq y \leq 1500\,\mu$m with a constant speed of 25 µm/s. Ten thousand cells were uniformly distributed in the space at $t = 0$. Each simulation was performed for at least four periods. An example of the traces of the mean positions in the $x$-direction (blue solid line) and $y$-direction (green solid line) is shown in *Figure 6C*. The cells effectively tracked the potassium gradient and their positions exhibited fluctuations in the direction of the gradient, similar to that observed experimentally by *Humphries et al., 2017*. Furthermore, a phase delay between the mean $x$-position and the potassium source was observed, indicating a temporal shift in response.

We simulated the taxis of cells under the oscillating source with $L_0 = 1$ mM and $T$ = 1, 2, 3, and 4 hr. Four typical traces are shown in *Figure 6D*, and the relation between the phase shift ($\Delta\phi$, as denoted in *Figure 6C*) and the driving period ($T$) is shown in *Figure 6E*. We found that the phase delay decreases significantly with the increase of driving period, a trend that is consistent with previous experimental observations in the biofilm (*Humphries et al., 2017*). Similar behavior was observed in frequency-dependent *E. coli* chemotaxis to spatially and temporally varying MeAsp source (*Zhu et al., 2012*). Furthermore, we also computed the lag time ($\Delta\phi \cdot T$) of the average $x$-position relative to the electric signal (potassium concentration). The relation between lag time and the driving period is shown in *Figure 6F*. For driving periods of 1, 2, 3, and 4 hr, the oscillation in the *E. coli* cell displacement lagged the potassium source by 11.7 ± 0.5, 14.7 ± 0.7, 14.8 ± 0.9, and 14.2 ± 1.0 min (mean ± SD), respectively. These values closely resemble the 9–35 min lag observed in biofilm experiments for *B. subtilis* and *P. aeruginosa* (*Humphries et al., 2017*). Notably, this result was not affected by the specific value of $L_0$ (*Figure 6—figure supplement 1*).

## Discussion

Potassium ions play a critical role in many physiological processes of bacteria. Bacteria in biofilms release potassium ions to their surroundings through ion channels on the cell membrane, thus influencing the behavior of surrounding bacteria and attracting distant free bacteria to swim toward them. However, the mechanism of bacterial sensing and response to potassium ions was unclear. Here, we found that *E. coli* senses the concentration change of extracellular potassium ions sensitively via the chemotaxis signaling pathway, and this chemotactic sensing is achieved through changes in intracellular pH.

We found that *E. coli* could quickly converge to the area with a higher concentration of potassium under a linear concentration profile. The measurements of the response of individual motors to stepwise addition and removal of 30 mM KCl via the bead assay demonstrated that the attractant response resulted from the chemotaxis signaling pathway instead of the possible PMF effect on the motor itself. We also demonstrated that the PMF remained unchanged when KCl was added or removed.

To directly measure the chemotactic response of *E. coli* to potassium ions, we measured the response to different concentrations of potassium via FRET between CheY-eYFP and CheZ-eCFP. After correcting the pH effects on the brightness of CFP and YFP proteins, we found that the chemotactic

response of wild-type *E. coli* to 30 mM KCl was 1.06±0.10 times that to 100 µM MeAsp. The response to 60 mM sucrose was −0.30 ± 0.05 times that to 100 µM MeAsp. This suggested that the attractant response to potassium was independent of the osmolality response. To quantitatively describe the chemotactic response to potassium, we systematically measured the dose–response curve and the step responses. We obtained a Hill coefficient of 0.88 ± 0.14 and a concentration of 0.33 ± 0.06 mM for the half-maximal response. This showed that *E. coli* had a very sensitive response to potassium ions and could sense weak changes in potassium concentration. We also found that *E. coli* adapts quickly to potassium concentration changes in the range of 0.01–100 mM, which promotes cell localization to the peak of a potassium concentration profile.

For other strong attractant stimuli such as high concentrations of MeAsp, the step response typically shows a low plateau before it adapts. However, in the case of potassium, the FRET signal does not typically display an obvious plateau following the stimuli. To observe the low plateau before adaptation, a saturating amount of attractant should be added in a stepwise manner. According to the dose–response curve we measured for potassium, a saturating amount of potassium would be close to 100 mM. In fact, there is a small segment of the low plateau in the step response to 30 mM KCl (*Figure 4C*). To observe more of this low plateau, we could have used a higher concentration of KCl. However, a stimulation higher than 30 mM KCl will induce substantial physiological changes in the cell, resulting in a significant decrease in fluorescence for both channels (*Figure 4—figure supplement 1*). Therefore, the range of KCl concentration that can be reliably applied in FRET measurements is limited.

While keeping the PMF unchanged, addition of potassium decreases the membrane potential and thus increases the intracellular pH. This was demonstrated by intracellular pH measurements. We also measured the chemotactic response of mutant strains expressing only Tsr or Tar to different concentrations of potassium. The Tsr-only strain responded to potassium as an attractant, whereas the Tar-only strain exhibited a biphasic response (a repellent-like followed by an attractant-like response). The different responses of Tar and Tsr to potassium were consist with their responses to changes in intracellular pH. We further demonstrated the biphasic response of the Tar-only strain to 40 mM sodium benzoate with pH of 7.0 that induced a decrease in intracellular pH. The differential responses of the Tsr and Tar receptors to potassium may be a strategy by which bacteria adjust their response to potassium at different growth stages during which bacteria express different ratios of Tsr to Tar receptors (*Kalinin et al., 2010*; *Yang and Sourjik, 2012*). For the growth stage used in our measurements here, the amount of Tsr was more than that of Tar in the wild-type strain (*Li and Hazelbauer, 2004*), so the response to potassium was dominated by Tsr.

The response of the Tar-only strain to potassium warrants further discussion (*Figure 5C*). This strain shows a repellent response to stepwise addition of low concentrations of potassium, specifically less than 10 mM. This is consistent with previous observations of the response of Tar to changes in intracellular pH (*Krikos et al., 1985*; *Umemura et al., 2002*). Interestingly, it exhibits a biphasic response to high potassium concentrations of 10 mM and above. This biphasic response might result from additional pH effects on the activity of intracellular enzymes such as CheRB and CheA (*Conley et al., 1994*), which may have a different timescale and response from the Tar receptor.

Based on the potassium sensing mechanism we established here, we performed stochastic simulation of bacterial taxis in a biofilm-produced potassium gradient, demonstrating that *E. coli* cells can be periodically attracted by the biofilm. Moreover, we observed a time delay between the positions of the cells and the electric signal of the biofilm. Specifically, we found that the oscillation of *E. coli* cell displacement lags the potassium source by 11.7 ± 0.5, 14.7 ± 0.7, 14.8 ± 0.9, and 14.2 ± 1.0 min (mean ± SD) for driving periods of 1, 2, 3, and 4 hr, respectively. Remarkably, these values closely resemble the 9–35 min lag observed in biofilm experiments for *B. subtilis* and *P. aeruginosa* (*Humphries et al., 2017*), despite the potential variations in the potassium sensing mechanisms among these bacterial species. The potassium sensing mechanism we established here contributed to our understanding of the complex dynamics of electrical signaling in biofilm environments.

# Materials and methods

**Key resources table**

| Reagent type (species) or resource | Designation | Source or reference | Identifiers | Additional information |
|---|---|---|---|---|
| Strain, strain background (*Escherichia coli*) | Wild-type AW405 | Howard Berg Lab *Armstrong et al., 1967* | HCB1 | Also known as AW405 |
| Strain, strain background (*Escherichia coli*) | Wild-type RP437 | Howard Berg Lab *Parkinson, 1978* | HCB33 | Also known as RP437 |
| Strain, strain background (*Escherichia coli*) | CheY** strain | Howard Berg Lab | HCB901 | $\Delta cheZ\ fliC$, Ptrc420 $cheY^{13DK106YW}$ |
| Strain, strain background (*Escherichia coli*) | RP437 with $\Delta cheY\ cheZ$ | Howard Berg Lab *Sourjik and Berg, 2002b* | HCB1288 | Also known as VS104; $\Delta cheY\ cheZ$ |
| Strain, strain background (*Escherichia coli*) | RP437 with $\Delta tar\ tsr\ tap\ trg\ aer\ cheY\ cheZ$ | Howard Berg Lab *Sourjik and Berg, 2002a* | HCB1414 | Also known as VS181; $\Delta tar\ tsr\ tap\ trg\ aer\ cheY\ cheZ$ |
| Strain, strain background (*Escherichia coli*) | RP437 with $\Delta fliC$ | This paper | JY26 | The *fliC* gene of strain RP437 was deleted; $\Delta fliC$ |
| Recombinant DNA reagent | pVS18 (plasmid) | Howard Berg Lab *Sourjik and Berg, 2002a* | pVS18 | CheY-eYFP |
| Recombinant DNA reagent | pVS88 (plasmid) | Howard Berg Lab *Sourjik and Berg, 2004* | pVS88 | CheY-eYFP and CheZ-eCFP |
| Recombinant DNA reagent | pBES38 (plasmid) | Howard Berg Lab | pBES38 | LacI$^q$ and FliC$^{sticky}$ |
| Recombinant DNA reagent | pKAF131 (plasmid) | Howard Berg Lab *Yuan et al., 2010* | pKAF131 | FliC$^{sticky}$ |
| Recombinant DNA reagent | pLC113 (plasmid) | Howard Berg Lab *Ames et al., 2002* | pLC113 | Tar |
| Recombinant DNA reagent | pPA114 (plasmid) | Howard Berg Lab *Ames et al., 2002* | pPA114 | Tsr |
| Recombinant DNA reagent | pTrc99a_pHluorin2 (plasmid) | This paper | pTrc99a_pHluorin2 | The gene *pHluorin2* was cloned into pTrc99a under an IPTG-inducible promoter. |
| Chemical compound, drug | Tryptone | Oxoid | CAT# LP0042B | |
| Chemical compound, drug | IPTG | Sigma-Aldrich | CAT# I6758 | |
| Chemical compound, drug | Lactic acid | Sigma-Aldrich | CAT# 252476 | |
| Software, algorithm | Custom script | *Zhang, 2024* | https://github.com/CZhang2023/2024_eLife_potassium | |

## Strains and plasmids

Strain HCB1 was derived from *E. coli* K12 strain AW405 (*Armstrong et al., 1967*). Strains JY26 ($\Delta fliC$), HCB33, HCB901 ($\Delta cheZ\ fliC$, Ptrc420 $cheY^{13DK106YW}$) (*Scharf et al., 1998*), HCB1288 ($\Delta cheY\ cheZ$) (*Sourjik and Berg, 2002b*), and HCB1414 ($\Delta tar\ tsr\ tap\ trg\ aer\ cheY\ cheZ$) (*Sourjik and Berg, 2002a*) were derived from *E. coli* K12 strain RP437 (*Parkinson, 1978*). The plasmid pKAF131 constitutively expresses FliC$^{sticky}$ (*Yuan et al., 2010*). The plasmid pBES38 constitutively expresses both LacI$^q$ and the sticky filament FliC$^{sticky}$ (*Scharf et al., 1998*). The promoters used for the constitutive expression of LacI$^q$ and FliC$^{sticky}$ were the I$^q$ promotor and the native promoter of *fliC*, respectively (*Scharf et al., 1998*). The FRET pair CheY-eYFP and CheZ-eCFP was expressed from plasmid pVS88 under

**Table 1.** Strains and plasmids used in this study.

| Strain | Plasmids | Assay |
|---|---|---|
| HCB1 | - | Microfluidic assay |
| JY26 | pKAF131 | Bead assay |
| HCB901 | pBES38 | Bead assay |
| | pVS88 | FRET assay |
| HCB1288 | pVS18 | FRET assay |
| | pVS88 | FRET assay |
| | pLC113, pVS88 | FRET assay |
| HCB1414 | pPA114, pVS88 | FRET assay |
| HCB33 | pTrc99a_pHluorin2 | Intracellular pH measurement |

an isopropyl-β-D-thiogalactoside (IPTG)-inducible promoter (*Sourjik and Berg, 2004*). The plasmid pVS18 was used to express CheY-eYFP under an IPTG-inducible promoter (*Sourjik and Berg, 2002a*). The plasmids pLC113 and pPA114 were used to express Tar and Tsr receptors under a salicylate-inducible promoter, respectively (*Ames et al., 2002*). The gene *pHluorin2* was cloned into pTrc99a under an IPTG-inducible promoter, yielding pTrc99a_pHluorin2. The strains and plasmids used in this study are shown in *Table 1*.

## Cell culture

All cells were grown at 33°C in 10 ml T-broth (1% tryptone and 0.5% NaCl) to an $OD_{600}$ between 0.45 and 0.50 with appropriate antibiotics and inducers, collected by centrifugation and washed twice with potassium-depleted motility buffer (10 mM $NaPO_4$, 0.1 mM ethylenediaminetetraacetic acid, 1 mM methionine, 10 mM lactic acid, pH 7.0). Cells of HCB1 were centrifuged for 6 min at 1200 × *g*. Cells of JY26-pKAF131 were grown with 25 μg/ml chloramphenicol. HCB901-pBES38 cells were grown with 100 μg/ml ampicillin and 23 μM IPTG. Both were centrifuged for 2 min at 4000 × *g*. Cells of HCB1288-pVS88 and HCB1414-pVS88 were grown with 100 μg/ml ampicillin and 100 μM IPTG in the dark. Cells of HCB1414-pLC113-pVS88 and HCB1414-pPA114-pVS88 were grown with 100 μg/ml ampicillin, 25 μg/ml chloramphenicol, 100 μM IPTG, and 1 μM salicylate in the dark. Cells of HCB33-pTrc99a_pHluorin2 were grown with 100 μg/ml ampicillin and 100 μM IPTG. Cells were stored at 4°C before use. All experiments were carried out at 23°C. The KCl solution used in this work was prepared in the potassium-depleted motility buffer.

## Microfluidics

We constructed a 100-μm-depth microfluidic device similar to that described previously (*Liu et al., 2022*; *Tian et al., 2021*). The design is shown in *Figure 1A*. There are two auxiliary channels and three primary channels. The width of the auxiliary channels is 100 μm, while the width of the primary channels is 400 μm. The source and sink channels were used to flow 100 mM KCl and potassium-depleted motility buffer, respectively, and the observing channel was used to observe the motion of cells. The auxiliary channels were filled with 2% agarose gel at a temperature of 68°C and cooled down. In our measurement, we kept the flow at a rate of 5 ml/min in both source and sink channels with a syringe pump (Pump-22; Harvard Apparatus). The potassium ions could diffuse from the source channel to the sink channel and form a linear gradient in the observing channel. Cells of HCB1 were sealed in the observing channel, and their movement was recorded by a fast CMOS camera (Flare 2M360-CL, IO Industries) equipped on a Nikon Ti-E microscope at a magnification of ×20. The chemotaxis migration coefficient, as an indicator of the mean cell position (*Kalinin et al., 2010*), was calculated as the average of $(x_i − x_c)/194$, where $x_i$ is the x-position of the *i*th cell, $x_c$ is the x-position of the center of the observing channel, and 194 μm is the half-width of the observed region.

## Bead assay

We sheared the sticky filaments of the washed cell suspensions by passing them 200 times between two syringes equipped with 23-gauge needles and connected by a 7-cm-long polyethylene tube (0.58 mm inside diameter, no. 0427411; Becton Dickinson), and condensed them into 300 μl in potassium-depleted motility buffer. To measure motor rotation, sheared cells were immobilized on a coverslip coated with poly-L-lysine (0.01%, P4707; Sigma, St. Louis, MO) assembled on a flow chamber (*Berg and Block, 1984*), and allowed to stand for 5 min. Then, 1-μm-diameter polystyrene beads (0.27%, no. 0731; Polysciences) were flowed to replace the suspension and attached to the sheared filament stubs by allowing it to stand for 4 min. The rotation of beads was recorded by a fast CMOS camera (Flare 2M360-CL, IO Industries) equipped on a Nikon Ti-E inverted phase-contrast microscope at a magnification of ×40. The CW bias and rotational speed were calculated with a 20-s time window and 1-s time slide.

## FRET assay

The experimental setup used in FRET measurements was the same as that described previously (*Zhang et al., 2018*). The FRET setup was based on a Nikon Ti-E microscope equipped with a 40× 0.60 NA objective. The illumination light was provided by a 130-W mercury lamp, attenuated by a factor of 1024 with neutral density filters, and passed through an excitation bandpass filter (FF02-438/24-25, Semrock) and a dichroic mirror (FF458-Di02−25x36, Semrock). The epifluorescent emission was split into cyan and yellow channels by a second dichroic mirror (FF509-FDi01−25x36, Semrock). The signals in the two channels were then filtered by two emission bandpass filters (FF01-483/32-25 and FF01-542/32-25, Semrock) and collected by two photon-counting photomultipliers (H7421-40, Hamamatsu, Hamamatsu City, Japan), respectively. Signals from the two photomultipliers were recorded at a sampling rate of 1 Hz using a data-acquisition card installed in a computer (USB-1901(G)−1020, ADlink, New Taipei, Taiwan).

The washed cell suspension was concentrated 45 times and flowed into a flow chamber equipped with a poly-L-lysine-coated coverslip, allowing it to stand for 20 min. Then, the chamber was maintained under a constant flow (500 $\mu$l/min) of potassium-depleted motility buffer by a syringe pump (Pump-22; Harvard Apparatus). The same flow was used to add and remove stimulus. All data analysis was performed using custom scripts in MATLAB 2018b (MathWorks). The FRET value was calculated as the ratio of YFP to CFP intensities, normalized by the pre-stimulus value. The impact of pH on the original signals for both CFP and YFP channels was corrected by

$$S_{pH-corrected}\left(L; t_{addition} < t < t_{removal}\right) = \frac{S_{orignal}\left(L; t_{addition} < t < t_{removal}\right)}{C_{pH}\left(L\right)}$$

where $S_{pH-corrected}\left(L; t_{addition} < t < t_{removal}\right)$ and $S_{orignal}\left(L; t_{addition} < t < t_{removal}\right)$ represent the pH-corrected and original PMT signal (CFP or YFP channel) from the moment of addition of $L$ mM KCl to the moment of its removal, respectively, and $C_{pH}\left(L\right)$ is the correction factor, which is the ratio of PMT signal post- to pre-KCl addition for the no-receptor mutant at $L$ mM KCl, for CFP or YFP channel as shown *Figure 5—figure supplement 3*. The pH-corrected FRET response is then calculated as the ratio of the pH-corrected YFP to the pH-corrected CFP, normalized by the pre-stimulus ratio.

For the YFP photobleaching assay, we measured the background of the CFP channel by recording the intensity of the CFP channel for the strain expressing only CheY-eYFP (HCB1288-pVS18). We monitored the response of the CFP signal to 30 mM KCl before and after bleaching YFP. The bleaching was carried out with a YFP filter set (C-FL YFP BP HYQ535, #41028, Chroma) under mercury lamp illumination without attenuation and sustained for 50 min. All these measurements were performed under the same conditions.

## Intracellular pH measurements

The washed cells of HCB33-pTrc99a_pHluorin2 were immobilized on a coverslip assembled on a flow chamber and coated with poly-L-lysine. Then, a constant flow (500 $\mu$l/min) was applied to the cells. The samples were illuminated periodically with two lasers. In each period, the laser with a wavelength of 405 nm was illuminated for 200 ms and stopped for 400 ms, and the laser with a wavelength of 488 nm was illuminated for another 200 ms and stopped for another 400 ms. The emission

light passed through the T495lp dichroic mirror and AT495lp longpass filter and was collected by an EMCCD (Andor DU897) equipped on a Nikon Ti-E inverted phase-contrast microscope at a magnification of ×100. The changes in intracellular pH could be characterized by the ratio of emitted fluorescence between the two excitation lasers.

## Simulation of bacterial taxis in an oscillating spatial gradient of potassium

In our simulation, cells were treated as self-propelled particles. They could swim smoothly (a run state) with a constant speed of 25 µm/s and with rotational diffusion (the rotational diffusion constant was 0.062 rad$^2$/s) (*Vladimirov et al., 2010*), or stop to reorient (a tumble state). The tumble angle $\theta$ was selected from the probability distribution $P(\theta) = 0.5 * (1 + \cos\theta) * \sin\theta \ (0 \leq \theta \leq \pi)$ (*Berg and Brown, 1972*; *Neumann et al., 2010*). A coarse-grained model of the chemotaxis signaling pathway (*Jiang et al., 2010*; *Tu et al., 2008*) was utilized to describe the sensing and adaptation of chemoreceptors to changes in extracellular potassium concentration:

$$a = \frac{1}{1 + \exp\left[N\left(\alpha(m - m_0) + \ln\frac{1 + L/K_{off}}{1 + L/K_{on}}\right)\right]}, \quad (3)$$

$$\frac{dm}{dt} = k_R(1 - a) - k_B a, \quad (4)$$

where $a$ is the kinase activity of the cluster of chemoreceptors, $m$ is the methylation level of receptors, and $L$ denotes the concentration of extracellular potassium. In the denominator on the right-hand side of *Equation 3*, the two terms within the parentheses of the exponential expression represent the methylation-dependent ($f_m$) and ligand-dependent ($f_l$) free energy, respectively. The parameters $N = 0.85$, $f_m = 0.84$, $K_{off} = 0.17 \, \text{mM}$, $K_{on} = 55.17 \, \text{mM}$ were determined by fitting the dose–response curve in *Figure 6B*. The values of the parameters $\alpha = -1.7$, $m_0 = 1.0$, $k_R = 0.005 \, s^{-1}$, $k_B = 0.010 \, s^{-1}$ were chosen to be the same as before (*Liu et al., 2022*).

The CheY-P concentration ($Yp$) is proportional to the kinase activity: $Yp = 7.86a$ (*Jiang et al., 2010*; *van Albada and Ten Wolde, 2009*), and is related to the motor CW bias ($B$) by (*Cluzel et al., 2000*):

$$B = \frac{Yp^{10.3}}{Yp^{10.3} + 3.1^{10.3}} \quad (5)$$

Then the switching rate from run to tumble and from tumble to run were determined by $B/0.11$ and $5 \, s^{-1}$, respectively (*Berg and Brown, 1972*; *He et al., 2016*).

In the simulation, cells swim in a square space with dimensions of $0 \leq x \leq 1500 \, \mu\text{m}$ and $0 \leq y \leq 1500 \, \mu\text{m}$. We implemented periodic boundary condition for the y-direction, and reflective boundaries for the x-direction. Ten thousand cells were uniformly distributed throughout the square space at $t = 0$. The time step is 0.01 s. They swam in a steady oscillating spatial gradient of potassium created by an oscillating source at $x = 0$. The simulation was performed for a minimum of four periods to observe and analyze the long-term behavior of the cells in response to the oscillating gradient. The simulated lag time decreases with the methylation rate $k_R$, but levels off at high values of $k_R$ (*Figure 6—figure supplement 2*).

## Acknowledgements

This work was supported by National Natural Science Foundation of China grants (11925406, 12090053, and 12104436), a grant from the Ministry of Science and Technology of China (2019YFA0709303), and the Fundamental Research Funds for the Central Universities (WK2030000026).

## Additional information

### Funding

| Funder | Grant reference number | Author |
|---|---|---|
| National Natural Science Foundation of China | 11925406 | Junhua Yuan |
| National Natural Science Foundation of China | 12090053 | Junhua Yuan |
| National Natural Science Foundation of China | 12104436 | Chi Zhang |
| Ministry of Science and Technology of the People's Republic of China | 2019YFA0709303 | Rongjing Zhang |
| Fundamental Research Funds for the Central Universities | WK2030000026 | Chi Zhang |

The funders had no role in study design, data collection, and interpretation, or the decision to submit the work for publication.

### Author contributions

Chi Zhang, Conceptualization, Data curation, Software, Formal analysis, Validation, Investigation, Visualization, Methodology, Writing – original draft, Writing – review and editing; Rongjing Zhang, Junhua Yuan, Conceptualization, Resources, Supervision, Funding acquisition, Investigation, Methodology, Writing – original draft, Project administration, Writing – review and editing

### Author ORCIDs

Chi Zhang ⓘ http://orcid.org/0000-0003-3154-2657
Rongjing Zhang ⓘ http://orcid.org/0009-0008-5519-6385
Junhua Yuan ⓘ http://orcid.org/0000-0002-6437-0655

Reviewer #1 (Public Review): https://doi.org/10.7554/eLife.91452.4.sa1
Author response https://doi.org/10.7554/eLife.91452.4.sa2

---

## Additional files

### Supplementary files

• MDAR checklist

### Data availability

All data generated or analyzed during this study are included in the manuscript and supporting files. Data analysis scripts were uploaded online (GitHub, copy archived at *Zhang, 2024*).

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

## Appendix 1

### The diffusion of potassium in semi-infinite liquid with a periodic source at $x = 0$

It has been found that the biofilm could lead to a periodically oscillated source of potassium. Here, we simplified this source as a concentration signal of a cosine function at $x = 0$. Then the potassium will diffuse in the region of $x > 0$ which could be described by

$$\frac{\partial C}{\partial t} = D\frac{\partial^2 C}{\partial x^2}, \text{ for } x > 0$$

with

$$C\left(x, t = 0\right) = 0,$$
$$C\left(x = 0, t\right) = \phi\left(t\right),$$

where $C$ is the concentration of potassium, $D$ is the diffusion constant of potassium in water, $L_0$ is the half maximum concentration of potassium, $T$ denotes the period of oscillation, and $\phi\left(t\right) = L_0\left(1 - \cos\left(2\pi t/T\right)\right)$.

The solution can be given by (**Carslaw and Jaeger, 1947**):

$$C\left(x, t\right) = \int_0^t \phi\left(\lambda\right)\frac{\partial F\left(x, t - \lambda\right)}{\partial t}d\lambda,$$

where

$$F\left(x, t - \lambda\right) = \frac{2}{\sqrt{\pi}}\int_{\frac{x}{2\sqrt{D\left(t - \lambda\right)}}}^{\infty} e^{-\xi^2}d\xi$$

Thus

$$C\left(x, t\right) = \frac{2}{\sqrt{\pi}}\int_{\frac{x}{2\sqrt{Dt}}}^{\infty} \phi\left(t - \frac{x^2}{4D\mu^2}\right) e^{-\mu^2}d\mu$$

where

$$\mu = \frac{x}{2\sqrt{D\left(t - \lambda\right)}}$$

In this case

$$C\left(x, t\right) = \frac{2}{\sqrt{\pi}}\int_{\frac{x}{2\sqrt{Dt}}}^{\infty} L_0\left(1 - \cos\left(\frac{2\pi}{T}\left(t - \frac{x^2}{4D\mu^2}\right)\right)\right) e^{-\mu^2}d\mu$$
$$= L_0\left[1 - \operatorname{erf}\left(\frac{x}{2\sqrt{Dt}}\right) - \frac{2}{\sqrt{\pi}}\int_{\frac{x}{2\sqrt{Dt}}}^{\infty} \cos\left(\frac{2\pi}{T}\left(t - \frac{x^2}{4D\mu^2}\right)\right) e^{-\mu^2}d\mu\right]$$

Since by a known definite integral

$$\frac{2}{\sqrt{\pi}}\int_0^{\infty} \cos\left(\frac{2\pi}{T}\left(t - \frac{x^2}{4D\mu^2}\right)\right) e^{-\mu^2}d\mu = \cos\left(\frac{2\pi}{T}t - \sqrt{\frac{\pi}{DT}}x\right) e^{-\sqrt{\frac{\pi}{DT}}x}$$

Then

$$C(x,t) = L_0 \left[ 1 - \cos\left(\frac{2\pi t}{T} - \sqrt{\frac{\pi}{DT}}x\right) \exp\left(-\sqrt{\frac{\pi}{DT}}x\right) \right]$$

$$-L_0 \left[ \mathrm{erf}\left(\frac{x}{2\sqrt{Dt}}\right) + \frac{2}{\sqrt{\pi}} \int_0^{\frac{x}{2\sqrt{Dt}}} \cos\left(\frac{2\pi}{T}\left(t - \frac{x^2}{4D\mu^2}\right)\right) e^{-\mu^2} d\mu \right]$$

The second term is a transient disturbance caused by starting the oscillations of source at $t = 0$, and it dies away as $t$ increases. In the steady state, for a finite value of $x$,

$$C(x,t) = L_0 \left[ 1 - \cos\left(\frac{2\pi t}{T} - \sqrt{\frac{\pi}{DT}}x\right) \exp\left(-\sqrt{\frac{\pi}{DT}}x\right) \right].$$

