## [Editor Report · eLife assessment]

In this **important** study, the authors report a novel measurement of the *Escherichia coli* chemotactic response and demonstrate that these bacteria display an attractant response to potassium, which is connected to intracellular pH level. The experimental evidence provided is **convincing** and the work will be of interest to microbiologists studying chemotaxis.

---

## [Referee Report · Reviewer #1 (Public Review)]

Summary:

This paper shows that *E. coli* exhibits a chemotactic response to potassium by measuring both the motor response (using a bead assay) and the intracellular signaling response (CheY phosporylation level via FRET) to step changes in potassium concentration. They find increase in potassium concentration induces a considerable attractant response, with amplitude comparable to aspartate, and cells can quickly adapt (and generally over-adapt). The authors propose that the mechanism for potassium response is through modifying intracellular pH; they find both that potassium modifies pH and other pH modifiers induce similar attractant responses. It is also shown, using Tar- and Tsr-only mutants, that these two chemoreceptors respond to potassium differently. Tsr has a standard attractant response, while Tar has a biphasic response (repellent-like then attractant-like). Finally, the authors use computer simulations to study the swimming response of cells to a periodic potassium signal secreted from a biofilm and find a phase delay that depends on the period of oscillation.

Strengths:

The finding that *E. coli* can sense and adapt to potassium signals and the connection to intracellular pH is quite interesting and this work should stimulate future experimental and theoretical studies regarding the microscopic mechanisms governing this response. The evidence (from both the bead assay and FRET) that potassium induces an attractant response is convincing, as is the proposed mechanism involving modification of intracellular pH. The updated manuscript controls for the impact of pH on the fluorescent protein brightness that can bias the measured FRET signal. After correction the response amplitude and sharpness (hill coefficient) are comparable to conventional chemoattractants (e.g. aspartate), indicating the general mechanisms underlying the response may be similar. The authors suggest that the biphasic response of Tar mutants may be due to pH influencing the activity of other enzymes (CheA, CheR or CheB), which will be an interesting direction for future study.

Weaknesses:

The measured response may be biased by adaptation, especially for weak potassium signals. For other attractant stimuli, the response typically shows a low plateau before it recovers (adapts). In the case of potassium, the FRET signal does not have an obvious plateau following the stimuli of small potassium concentrations, perhaps due to the faster adaptation compared to other chemoattractants. It is possible cells have already partially adapted when the response reaches its minimum, so the measured response may be a slight underestimate of the true response. Mutants without adaptation enzymes appear to be sensitive to potassium only at much larger concentrations, where the pH significantly disrupts the FRET signal; more accurate measurements would require the development of new mutants and/or measurement techniques.

Note added after the second revision: The authors made a reasonable argument regarding the effects of adaptation, which were estimated to be small.

---

## [Author Response]

The following is the authors’ response to the previous reviews.

**eLife assessment**
In this important study, the authors report a novel measurement of the *Escherichia coli* chemotactic response and demonstrate that these bacteria display an attractant response to potassium, which is connected to intracellular pH level. Whilst the experiments are mostly convincing, there are some confounders regards pH changes and fluorescent proteins that remain to be addressed.
**Public Reviews:**

**Reviewer #1 (Public Review):**
Summary:This paper shows that *E. coli* exhibits a chemotactic response to potassium by measuring both the motor response (using a bead assay) and the intracellular signaling response (CheY phosporylation level via FRET) to step changes in potassium concentration. They find increase in potassium concentration induces a considerable attractant response, with amplitude comparable to aspartate, and cells can quickly adapt (and generally over-adapt). The authors propose that the mechanism for potassium response is through modifying intracellular pH; they find both that potassium modifies pH and other pH modifiers induce similar attractant responses. It is also shown, using Tar- and Tsr-only mutants, that these two chemoreceptors respond to potassium differently. Tsr has a standard attractant response, while Tar has a biphasic response (repellent-like then attractant-like). Finally, the authors use computer simulations to study the swimming response of cells to a periodic potassium signal secreted from a biofilm and find a phase delay that depends on the period of oscillation.Strengths:The finding that *E. coli* can sense and adapt to potassium signals and the connection to intracellular pH is quite interesting and this work should stimulate future experimental and theoretical studies regarding the microscopic mechanisms governing this response. The evidence (from both the bead assay and FRET) that potassium induces an attractant response is convincing, as is the proposed mechanism involving modification of intracellular pH. The updated manuscript controls for the impact of pH on the fluorescent protein brightness that can bias the measured FRET signal. After correction the response amplitude and sharpness (hill coefficient) are comparable to conventional chemoattractants (e.g. aspartate), indicating the general mechanisms underlying the response may be similar. The authors suggest that the biphasic response of Tar mutants may be due to pH influencing the activity of other enzymes (CheA, CheR or CheB), which will be an interesting direction for future study.Weaknesses:The measured response may be biased by adaptation, especially for weak potassium signals. For other attractant stimuli, the response typically shows a low plateau before it recovers (adapts). In the case of potassium, the FRET signal does not have an obvious plateau following the stimuli of small potassium concentrations, perhaps due to the faster adaptation compared to other chemoattractants. It is possible cells have already partially adapted when the response reaches its minimum, so the measured response may be a slight underestimate of the true response. Mutants without adaptation enzymes appear to be sensitive to potassium only at much larger concentrations, where the pH significantly disrupts the FRET signal; more accurate measurements would require development of new mutants and/or measurement techniques.

We acknowledge and appreciate the reviewer's concerns regarding the potential impact of adaptation on the measured response magnitude. We have estimated the effect of adaptation on the measured response magnitude. The half-time of adaptation at 30 mM KCl was measured to be approximately 80 s, corresponding to a time constant of t = 80/ln(2) = 115.4 s, which is significantly longer than the time required for medium exchange in the flow chamber (less than 10 s). Consequently, the relative effect of adaptation on the measured response magnitude should be less than 1-exp(-10/t) = 8.3%. Even for the fastest adaptation (at the lowest KCl concentration) we measured, the effect should be less than 20%, which is within experimental uncertainties. Nevertheless, we agree that developing new techniques to measure the dose-response curve more precisely would be beneficial.

**Reviewer #2 (Public Review):**
Zhang et al investigated the biophysical mechanism of potassium-mediated chemotactic behavior in E coli. Previously, it was reported by Humphries et al that the potassium waves from oscillating B subtilis biofilm attract P aeruginosa through chemotactic behavior of motile P aeruginosa cells. It was proposed that K+ waves alter PMF of P aeruginosa. However, the mechanism was this behaviour was not elusive. In this study, Zhang et al demonstrated that motile E coli cells accumulate in regions of high potassium levels. They found that this behavior is likely resulting from the chemotaxis signalling pathway, mediated by an elevation of intracellular pH. Overall, a solid body of evidence is provided to support the claims. However, the impacts of pH on the fluorescence proteins need to be better evaluated. In its current form, the evidence is insufficient to say that the fluoresce intensity ratio results from FRET. It may well be an artefact of pH change.The authors now carefully evaluated the impact of pH on their FRET sensor by examining the YFP and CFP fluorescence with no-receptor mutant. The authors used this data to correct the impact of pH on their FRET sensor. This is an improvement, but the mathematical operation of this correction needs clarification. This is particularly important because, looking at the data, it is not fully convincing if the correction was done properly. For instance, 3mM KCl gives 0.98 FRET signal both in Fig3 and FigS4, but there is almost no difference between blue and red lines in Fig 3. FigS4 is very informative, but it does not address the concern raised by both reviewers that FRET reporter may not be a reliable tool here due to pH change.

We apologize for not making the correction process clear. We corrected the impact of pH on the original signals for both CFP and YFP channels bySpH− corrected (L;taddition <t<tremoval )=Sorignal (L;taddition <t<tremoval )CpH(L)

where SpH− corrected (L;taddition <t<tremoval ) and Sorignal (L;taddition <t<tremoval) represent the pH-corrected and original PMT signal (CFP or YFP channel) from the moment of addition of L mM KCl to the moment of its removal, respectively, and is the correction factor, which is the ratio of PMT signal post- to pre-KCl addition for the no-receptor mutant at L mM KCl, for CFP or YFP channel as shown Fig. S5. The pH-corrected FRET response is then calculated as the ratio of the pH-corrected YFP to the pH-corrected CFP signals, normalized by the pre-stimulus ratio.

As shown in Author response image1, which represents the same data as Fig. 3A and Fig. S5A, the original normalized FRET responses to 3 mM KCl are 0.967 for the wild-type strain (Fig. 3) and 0.981 for the no-receptor strain (Fig. S5). The standard deviation of the FRET values under steady-state conditions is 0.003. Thus, the difference in responses between the wild-type and no-receptor strains is significant and clearly exceeds the standard deviation. The pH correction factors CpH at 3 mM KCl are 1.004 for the YFP signal and 1.016 for the CFP signal. Consequently, the pH-corrected FRET responses are 0.967´1.016/1.004=0.979 for the wild-type and 0.981´1.016/1.004=0.993 for the no-receptor strain. The reason the pH-corrected FRET response for the no-receptor strain is 0.993 instead of the expected 1.000 is that this value represents the lowest observed response rather than the average value for the FRET response.

The detailed mathematical operation for correcting the pH impact has now been included in the “FRET assay” section of Materials and Methods.

**Author response image 1. sa2fig1:** Chemotactic response of the wild-type strain (**A**, HCB1288-pVS88) and the no-receptor strain (**B**, HCB1414-pVS88) to stepwise addition and removal of KCl. The blue solid line denotes the original normalized signal. Downward and upward arrows indicate the time points of addition and removal of 3 mM KCl, respectively. The horizontal red dashed line denotes the original normalized FRET response value to 3 mM KCl.

The authors show the FRET data with both KCl and K2SO4, concluding that the chemotactic response mainly resulted from potassium ions. However, this was only measured by FRET. It would be more convincing if the motility assay in Fig1 is also performed with K2SO4. The authors did not address this point. In light of complications associated with the use of the FRET sensor, this experiment is more important.

We thank the reviewer for the suggestion. We agree that additional confirmation with a motility assay is important. To address this, we have now measured the response of the motor rotational signal to 15 mM K2SO4 using the bead assay and compared it with the response to 30 mM KCl. The results are shown in Fig. S2. The response of motor CW bias to 15 mM K2SO4 exhibited an attractant response, characterized by a decreased CW bias upon the addition of K2SO4, followed by an over-adaptation that is qualitatively similar to the response to 30 mM KCl. However, there were notable differences in the adaptation time and the presence of an overshoot. Specifically, the adaptation time to K2SO4 was shorter compared to that for KCl, and there was a notable overshoot in the CW bias during the adaptation phase. These differences may have resulted from the weaker response to K2SO4 (Fig. S1B) and additional modifications due to CysZ-mediated cellular uptake of sulfate (Zhang et al., Biochimica et Biophysica Acta 1838,1809–1816 (2014)). The faster adaptation and overshoot complicated the chemotactic drift in the microfluidic assay as in Fig. 1, such that we were unable to observe a noticeable drift in a K2SO4 gradient under the same experimental conditions used for the KCl gradient.

The response of motor rotational signal to 15 mM K2SO4 has been added to Fig. S2.

**Recommendations for the authors:**

**Reviewer #1 (Recommendations For The Authors):**
(1) The response curve and adaptation level/time in the main text (Fig. 4) should be replaced by the corrected counterparts (currently in Fig. S5). The current version is especially confusing because Fig. 6 shows the corrected response, but the difference from Fig. 4 is not mentioned.

We thank the reviewer for the suggestion. We have now merged the results of the original Fig. S5 into Fig. 4.

a. The discussion of the uncorrected response with small hill coefficient and potentially negative cooperativity was left in the text (lines 223-234), but the new measurements show this is not true for the actual response. This should be removed or significantly rephrased.

We thank the reviewer for the suggestion. We have now removed the statement about potentially negative cooperativity and added the corrected results for the actual response.

(2) It may be helpful to restate the definition of f_m in the methods (near Eq. 3-4).

Thank you for the suggestion. We have now restated the definition of fm and fL below Eq. 3-4: “In the denominator on the right-hand side of Eq. 3, the two terms within the parentheses of exponential expression represent the methylation-dependent (fm) and ligand-dependent (fL) free energy, respectively.”